Assembly and evolutionary analysis of the complete mitochondrial genome of Trichosanthes kirilowii, a traditional Chinese medicinal plant

Jiang Zhuanzhuan zzjiang@whu.edu.cn 1 2
Chen Yuhan 1 2
Zhang Xingyu 1 2
Meng Fansong 1 2
Chen Jinli 1 2
Cheng Xu 1 2
1 Anqing Normal University , Anqing , Anhui , China
2 Provincial Key Laboratory of the Biodiversity Study and Ecology Conservation in Southwest Anhui , Anqing , Anhui , China
Zhang Lin
Electronic publication date: 2024 Jul 18
Publication date: 2024
Volume: 12
Electronic Location ID: e17747
Received 2024 Apr 30; Accepted 2024 Jun 24
Copyright: ©2024 Jiang et al.
Copyright year: 2024
Copyright holder: Jiang et al.
License: This is an open access article distributed under the terms of the Creative Commons Attribution License, which permits unrestricted use, distribution, reproduction and adaptation in any medium and for any purpose provided that it is properly attributed. For attribution, the original author(s), title, publication source (PeerJ) and either DOI or URL of the article must be cited.
License URL: https://creativecommons.org/licenses/by/4.0/

Keywords: Trichosanthes kirilowii, Mitochondrial genome, Multi-branched molecule, RNA editing, Gene transfer

Funding: The State Key Laboratory of Hybrid Rice Program (Wuhan University) KF202204 Scientific Research Foundation of the Higher Education Institutions of Anhui Province 2022AH051034 2022AH051048 2023 University Excellent Talent Training Project JWFX2023027 The research was financially supported by the State Key Laboratory of Hybrid Rice Program (Wuhan University) (KF202204), the Scientific Research Foundation of the Higher Education Institutions of Anhui Province (2022AH051034, 2022AH051048), and the 2023 University Excellent Talent Training Project (JWFX2023027). The funders did not participate in study design, data collection and analysis, decision to publish, or manuscript preparation. The funders had no role in study design, data collection and analysis, decision to publish, or preparation of the manuscript.

==============================
Trichosanthes kirilowii (T. kirilowii) is a valuable plant used for both medicinal and edible purposes. It belongs to the Cucurbitaceae family. However, its phylogenetic position and relatives have been difficult to accurately determine due to the lack of mitochondrial genomic information. This limitation has been an obstacle to the potential applications of T. kirilowii in various fields. To address this issue, Illumina and Nanopore HiFi sequencing were used to assemble the mitogenome of T. kirilowii into two circular molecules with sizes of 245,700 bp and 107,049 bp, forming a unique multi-branched structure. The mitogenome contains 61 genes, including 38 protein-coding genes (PCGs), 20 tRNAs, and three rRNAs. Within the 38 PCGs of the T. kirilowii mitochondrial genome, 518 potential RNA editing sites were identified. The study also revealed the presence of 15 homologous fragments that span both the chloroplast and mitochondrial genomes. The phylogenetic analysis strongly supports that T. kirilowii belongs to the Cucurbitaceae family and is closely related to Luffa. Collinearity analysis of five Cucurbitaceae mitogenomes shows a high degree of structural variability. Interestingly, four genes, namely atp1, ccmFC, ccmFN, and matR, played significant roles in the evolution of T. kirilowii through selection pressure analysis. The comparative analysis of the T. kirilowii mitogenome not only sheds light on its functional and structural features but also provides essential information for genetic studies of the genus of Cucurbitaceae.

Introduction

Cucurbitaceae belongs to the order Cucurbit ales, encompassing 97 genera and approximately 1,000 species worldwide (Ma et al., 2022). Among these, 55 species have been cultivated and utilized globally, demonstrating potential economic value (Guo et al., 2020). These include common vegetables such as cucumber, winter melon, pumpkin, loofah, and bitter melon, as well as fruits like watermelon, melon, and cantaloupe (Chomicki, Schaefer & Renner, 2020). In recent years, genome sequencing of economically important crops such as cucumber, watermelon, pumpkin, Momordica, and monk fruit has been completed (Li et al., 2019; Luo et al., 2023; Yang et al., 2023c). This advancement has significantly advanced research on the economic traits of Cucurbitaceae plants. Trichosanthes kirilowii Maxim is a perennial climbing vine belonging to the genus Trichosanthes of the Cucurbitaceae family. T. kirilowii is highly esteemed for its herbaceous medicinal properties and has a long history of traditional use in Chinese medicine (Zhang et al., 2019). The fruit parts are primarily employed for the treatment of diseases in clinical settings, while the roots are rich in protein (Chen et al., 2018; Xu et al., 2012). Additionally, the leaves, stems, fruit pulp and root bark are often important components as well (Yu et al., 2018). However, the current phylogenetic studies of Cucurbitaceae primarily rely on chloroplast genes and a limited number of nuclear gene molecular markers (Kousar & Park, 2023; Park et al., 2021; Renner, Schaefer & Kocyan, 2007; Wang et al., 2022a). As a result, the relationships between the main branches and genera within the family are not clearly understood. In particular, the relationships between the earliest branches and recently differentiated branches remain uncertain. These issues require further investigation and resolution to provide a more comprehensive understanding of Cucurbitaceae phylogeny.

Mitochondria, an organelle that plays an indispensable role in eukaryotic cells, is essential for energy metabolism and information transmission. Mitochondria in higher plants are the predominant presence among all known higher organelles. They possess their genetic material, but their genetic system is still reliant on the genetic system of the nucleus, thus forming a partially semi-autonomous genetic system (Forner et al., 2022; Møller, Rasmusson & Van Aken, 2021). Compared to the plastid genome, the mitochondrial genome of flowering plants (angiosperms) exhibits significant variability in their genetic structure and size (Yue et al., 2023; Zhang et al., 2023a; Zhang et al., 2023b). Due to this reason, our understanding of the mitochondrial genome of plants lags far behind that of the chloroplast genome. There is a lack of data on the mitochondrial genome of plants compared to the abundance of information available on the chloroplast genome. As of April 2024, the NCBI Organelle Genome Database (https://www.ncbi.nlm.nih.gov/genome/) has only released 602 plant mitogenomes, while over 10,479 chloroplast genomes have been made available.

Due to complex structural variations and long periods of repetitive DNA, genome assembly for angiosperms is challenging (Mower, 2020; Xia et al., 2023). Recent studies have revealed that the structure of angiosperm mitochondria shows high variability, repetitive DNA sequences, low gene density, and a low nucleotide substitution rate (Wu et al., 2020). The composition of the mitochondrial genome, particularly protein-coding, RNA, and trans-gene content, is relatively stable (Sun et al., 2022; Wang et al., 2022b; Zhang et al., 2023a). However, the conservation levels of these gene sequences can vary widely depending on the taxa (Sun et al., 2022). In contrast, the rate of gene rearrangement in plant mitochondrial genomes is significantly higher than that observed in plastids and animal genomes (Sullivan et al., 2020). The presence of variously sized repeats in the plant mitochondrial genome could potentially result in a higher frequency of genome rearrangement, and even facilitate the emergence of alternative genomic forms through recombination (Dong et al., 2018; Sun et al., 2022; Yang et al., 2022a; Yang et al., 2022b). It is important to note that mitochondrial genome rearrangement is a common occurrence, and the order of genes is rarely conserved among closely related species. Although the size of plant mitochondrial genomes varies, they consistently demonstrate a conserved GC content (Lai et al., 2022; Martins et al., 2019). The stability of evolved species is reflected in the GC content of higher plant mitochondrial genomes. The substantial increase in the number of repetitions is a contributing factor to the extensive expansion of mitochondrial genomes (Xiong et al., 2022). The acquisition of exogenous sequences from nuclear and plastid genomes serves as a significant source of non-coding sequences, potentially leading to the enlargement of mitochondrial genome size in angiosperms (Choi & Park, 2021; O’Conner & Li, 2020; Yang et al., 2023b).

RNA editing is prevalent in mitochondrial genomes and represents a crucial step for gene expression in plant mitochondrial genomes (Edera, Gandini & Sanchez-Puerta, 2018). RNA editing enhances the homology of mitochondrial proteins across different species, resulting in increased conservation of the encoded protein and improved expression (Small et al., 2019). Currently, the analysis of RNA editing, specifically the conversion of cytidine (C) to uridine (U), is widely conducted in plant mitochondria and chloroplasts (Yang et al., 2023a). In some plant taxa, a “reverse” U-to-C editing process has been observed (Edera et al., 2021). The biological significance of RNA editing lies in its capacity to rectify and regulate translation, as well as expand genetic information within an organism (Yang et al., 2023a).

This research utilized Illumina and Nanopore technologies to obtain the full sequences of T. kirilowii’s mitochondrial genome. The main objective of the study was to enhance our understanding of its morphological characteristics and genetic information at a molecular level. Additionally, this research aims to clarify the previous relatives of each species within the cucurbit family and to extensively explore the genetic resources of other medicinal plants within the cucurbit family, including T. kirilowii.

Material and Methods

Plant materials and growth condition

The Provincial Key Laboratory of the Biodiversity Study and Ecology Conservation in Southwest Anhui provided the T. kirilowii seeds. These seeds were soaked for 24 h before being transferred to a growth chamber set at a temperature of 28 °C. After germination, the plants were moved to nutrient-rich soil and cultivated in a greenhouse. The greenhouse is set at a temperature of 25 °C with a light-dark photoperiod cycle of 16 h of light and 8 h of darkness.

DNA extraction and sequencing

Genomic DNA was extracted from the young leaves of 14-day-old T. kirilowii plants using the CTAB method. The quality and concentration of the DNA were assessed using 1% agarose gel electrophoresis, the NanoPhotometer (IMPLEN, Westlake Village, CA, USA), and the Qubit 3.0 Fluorometer (Life Technologies, Carlsbad, CA, USA). To sequence the mitochondrial and chloroplast genomes of T. kirilowii, the Illumina and Nanopore methodologies were employed. Paired-end libraries with an insert size of 300 bp were prepared and sequenced using the Illumina HiSeq2500 platform. Low-quality reads were filtered out using the SOAPnuke (v2.1.4) tool to ensure data quality. The Oxford Nanopore sequencing libraries were prepared using established protocols with the ligation kit. The purified library was loaded onto primed R9.4 Spot-on Flow Cells and sequenced using a PromethION sequencer (Oxford Nanopore Technologies, Oxford UK) for 48-72 h. The raw data was base-called using the Oxford Nanopore GUPPY software.

Assembly and annotation of T. kirilowii

The mitochondrial genome of T. kirilowii was assembled using a combination of long-read and short-read data. Long-read data were used for the initial assembly, while short-read data were employed for precise correction. To achieve this, we used Flye software (Kolmogorov et al., 2019) with default settings to directly assemble the long-read sequencing data. The output of this process was graphical assembly outputs in GFA format. We utilized the assembled fasta format contigs to construct a library using makeblastdb. We then employed the BLASTn program to identify contigs that harbored the mitochondrial genome. In this study, we used conserved plant mitochondrial genes in Arabidopsis thaliana (L.) Heynh as query sequences. To visualize the GFA file, we used the Bandage software (v0.8.1) (Wick et al., 2015). We screened the mitochondrial contigs based on BLASTn results to obtain the draft mitochondrial genome of T. kirilowii. We used the bwa software (v0.7.17) (Li & Durbin, 2009) to align the long-reads and short-read data to the mitochondrial contigs, filtering and exporting the aligned mitochondrial reads separately for subsequent mixed assembly. Finally, we used Unicycler (Wick et al., 2017) with default parameters for the assembly process, resulting in the final T. kirilowii mitochondrial genome.

Two reference genomes for protein-coding genes of the mitochondrial genome were chosen for this study: Arabidopsis thaliana (NC_037304) and Liriodendron tulipifera L. (NC_021152.1). The mitochondrial genome was annotated using Geseq software (v2.03) (Tillich et al., 2017), which included tRNAs with tRNAscan-SE software (v.2.0.11) (Lowe & Sean, 1997), and rRNAs with BLASTN software (v2.13.0) (Chen et al., 2015). Any errors in the annotation were manually corrected using Apollo software (v1.11.8) (Lewis et al., 2002).

Codon usage bias, repeat elements, and RNA editing site analysis

The Phylosuite software (v1.1.16) developed by Zhang et al. (2020) in 2020 was utilized to analyze the genome and extract protein-coding sequences. Additionally, the Mega software (v7.0) developed by Kumar, Stecher & Tamura (2016) was employed to calculate the RSCU values and study the codon bias in the mitochondrial genome. We also used the Codonw software (v1.4.2) to further examine the codon bias of protein-coding genes in the mitochondrial genome, which involved the calculation of various parameters such as T3s, C3s, A3s, G3s, CAL, FOP, CBI, and ENC.

The microsatellite sequences and tandem repeat sequences were identified using MISA (v2.1) (Beier et al., 2017), agency (v4.09) (Benson, 1999), and the REPuter web server (Kurtz et al., 2001). The results were visualized using GraphPad Prism (version 8.0) and the Rcircos package (v0.69-9) (Zhang, Meltzer & Davis, 2013).

The sequences of all protein-coding genes (PCGs) encoded by the mitochondrial genome of T. kirilowii were utilized as input files for Deepred-mt (Edera et al., 2021), a convolutional neural network-based tool with high accuracy in predicting RNA editing sites from C to U in mitochondrial PCGs. Only results with probability values exceeding 0.9 were retained.

Transfer fragment and synteny analysis

The process of assembling the chloroplast genome included using GetOrganelle software (v1.7.7.0) (Jin et al., 2020), annotating it with CPGAVAS2 software (Shi et al., 2019), and then correcting the annotation using CPGView software (Liu et al., 2023). Homologous fragments were analyzed using BLASTN software (v2.13.0) (Chen et al., 2015), and their results were visualized using the Rcircos package (v0.69-9) (Zhang, Meltzer & Davis, 2013). For synteny analysis, the BLAST program was used to perform pairwise comparisons of each mitochondrial genome to obtain BLASTN results. Homologous sequences with a length exceeding 500 bp were selectively retained as conserved collinear blocks for generating Multiple Synteny plots.

Organellar phylogenetical inference

A total of 37 complete mitochondrial genome sequences were retrieved from the National Center for Biotechnology Information (NCBI) database, representing four distinct orders (Cucurbitales, Rosales, Fagales, and Fabales). These included Cucumis sativus L. (NC_016004), Cucumis hystrix Chakr. (OK326879), Cucumis melo subsp. agrestis (Naudin) Pangalo. (MW854328), Cucurbita pepo L. (NC_014050.1), Cucurbita maxima Duchesne ex Lam. (OL350846.1), Citrullus lanatus (Thunb.) Matsum. & Nakai. (NC_014043.1), Lagenaria siceraria (Molina) Standl. (OR680814.1), Luffa aegyptiaca Mill. (OR346920), Luffa acutangula (L.) Roxb. (NC_050067.1), Herpetospermum pedunculosum (Ser.) C. B. Clarke. (MW497575.1), Momordica charantia L. (NC_077563.1), Fragaria tibetica Staudt & Dickoré. (NC_062832.1), Fragaria iturupensis Staudt. (NC_062833.1), Potentilla micrantha Ramond ex DC. (NC_062588.1), Rosa rugosa Thunb. (NC_065237.1), Rosa chinensis Jacq. (NC_065236.1), Geum urbanum L. (NC_065221.1), Rubus chingii Hu. (NC_065238.1), Sorbus aucuparia Poir. (NC_052880.1), Photinia serratifolia (Desf.) Kalkman. (NC_065220.1), Eriobotrya japonica (Thunb.) Lindl. (NC_045228.1), Sorbus torminalis (NC_052879.1), Pyrus betulifolia Bunge. (NC_054332.1), Malus x domestica (Suckow) Borkh. (NC_018554.1), Malus baccata (L.) Borkh. (NC_065224.1), Prunus sibirica L. (NC_065234.1), Prunus mume Siebold & Zucc. (NC_065232.1), Ziziphus mauritiana Lam. (NC_068745.1), Ziziphus jujuba (L.) Lam. (NC_029809.1), Cannabis sativa L. (NC_029855.1), Morus notabilis C. K. Schneid. (NC_041177.1), Quercus acutissima Carruth. (MZ636519.1), Lithocarpus litseifolius (Hance) Chun. (NC_065018.1), Fagus sylvatica L. (NC_050960.1), Juglans mandshurica Maxim. (MZ900993). The mitochondrial genomes of Vigna radiata (L.) R. Wilczek. (NC_015121.1) and Glycine max (L.) Merr. (NC_020455.1) were used as outgroups. 26 conserved genes (atp 1, atp 4, atp 6, atp 8, ccm B, ccm C, ccm FC, ccm FN, cob, cox 3, mat R, mtt B, nad 1, nad 2, nad 3, nad 4, nad 4L, nad 5, nad 6, nad 7, nad 9, rpl 2, rpl 16, rps 3, rps 4, and rps 19) were extracted using PhyloSuite software (Zhang et al., 2020), followed by conducting multiple sequence alignment analyses with MAFFT software (v7.505) (Katoh & Standley, 2013). The phylogenetic analysis was performed using IQ-TREE software (v1.6.12) (Nguyen et al., 2015) with maximum likelihood approache, and the results were visualized using ITOL software (v6) (Letunic & Bork, 2019).

Molecular evolutionary analysis

We analyzed the non-synonymous/synonymous mutation ratio (Ka/Ks) for the 31 shared protein-coding genes (PCGs) in mitochondrial DNA (mtDNA) from five different Cucurbitaceae species. To do this, we used Ka/Ks Calculator software (version 2.0). For the calculation of nucleic acid variability, we used DnaSP 6 (Rozas et al., 2017), with all parameters set to their default values. GraphPad Prism (version 8.0) was used to create charts, and Adobe Illustrator (version 2019) was used to enhance visual aesthetics.

Results

Multibranched conformations and genomic features of the T. kirilowii mtDNA

The mitochondrial genome draft assembled using long-read data was visualized using Bandage software (v0.8.1) (Wick et al., 2015). This draft was made up of six nodes, which were represented by six contigs obtained through assembly. The lengths of the contigs are as follows: contig 1 (148,018 bp), contig 2 (97,169 bp), contig 3 (58,416 bp), contig 4 (48,120 bp), contig 5 (277 bp), and contig 6 (236 bp) (Fig. 1A). Each node has a name assigned to it. If two nodes are connected by black lines, it indicates an overlap between the corresponding sequences. The red nodes represent potential repetitive elements that may occur multiple times in the genome. Figure 1 shows the sequences of two circular contigs obtained by resolving the branching nodes caused by these repetitive elements (red nodes) using long-read data.

Figure 1 (A–B) The assembly result of the mitochondrial genome of T. kirilowii.

The mitochondrial genome of T. kirilowii exhibits a complex multi-branched structure. By employing Nanopore data to exclude repetitive regions, we successfully obtained two primary circular contigs with a combined length of 352,749 bp: chromosome 1 spanning 245,700 bp, and chromosome 2 spanning 107,049 bp (Fig. 2). The overall GC content was determined to be 45.39%, with chromosome 1 exhibiting a slightly lower GC content of 45.29% and chromosome 2 showing a slightly higher GC content of 45.61% (Table 1). The mitochondrial genome of T. kirilowii was comprehensively annotated, revealing a total of 38 distinct protein-coding genes, comprising 24 essential mitochondrial core genes and 14 unique non-core genes. Additionally, the annotation identified 20 tRNA genes (including three multi-copy tRNAs) and three rRNA genes (Table 2).

Figure 2 The assembly result of the mitochondrial genome of T. kirilowii.

Photo credit: Xu Cheng.

Table 1 Accession number for the mitochondrial genome of T. kirilowii.

NCBI accession number	Contigs	Type	Length	GC content	
PP625757.1	Chromosome 1-2	Branched	352,749 bp	45.39%	
	Chromosome 1	Circular	245,700 bp	45.29%	
	Chromosome 2	Circular	107,049 bp	45.61%	

Table 2 Mitochondrial protein-coding genes of T. kirilowii.

Group of genes	Name of genes	
ATP synthase	atp 1, atp 4, atp 6, atp 8, atp 9	
NADH dehydrogenase	nad 1, nad 2, nad 3, nad 4, nad 4L, nad 5, nad 6, nad 7, nad 9	
Cytochrome b	cob	
Cytochrome c biogenesis	ccm B, ccm C, ccm FC, ccm FN	
Cytochrome c oxidase	cox 1, cox 2, cox 3	
Maturases	mat R	
Protein transport subunit	mtt B	
Ribosomal protein large subunit	rpl 2, rpl 5, rpl 10, rpl 16	
Ribosomal protein small subunit	rps 1, rps 3, rps 4, rps 7, rps 10, rps 12, rps 13, rps 19	
Succinate dehydrogenase	sdh 3, sdh 4	
Ribosome RNA	rrn 5, rrn 18, rrn 26	
Transfer RNA	trn A-UGC, trn C-GCA (×2), trn D-GUC, trn E-UUC, trn F-GAA, trn fM-CAU, trn G-GCC, trn H-GUG, trn I-CAU (×2), trn K-UUU, trn L-CAA, trn L-UAA, trn M-CAU, trn N-GUU, trn P-UGG (×2), trn Q-UUG, trn S-GCU, trn S-UGA, trn W-CCA, trn Y-GUA	

The core genes consist of five ATP synthase genes (atp 1, atp 4, atp 6, atp 8, and atp 9), nine NADH dehydrogenase genes (nad 1, nad 2, nad 3, nad 4, nad 4L, nad 5, nad 6, nad 7, and nad 9), four cytochrome C biogenesis genes (ccm B, ccm C, ccm FC and ccm FN), three cytochrome C oxidase genes (cox 1, cox 2 and cox 3), one membrane transport protein gene (mtt B), one maturase gene (mat R) and one panthenol-cytochrome C reductase gene (cob). The non-core genes identified in this study encompassed four ribosomal large subunit genes (rpl 2, rpl 5, rpl 10, rpl 16), eight small subunit genes (rps 1, rps 3, rps 4, rps 7, rps 10, rps 12, rps 13, and rps 19), as well as two succinate dehydrogenase genes (sdh 3 and sdh 4) (Table 2).

Analysis of codon usage bias

The codon bias analysis was conducted on 38 unique protein-coding genes (PCGs) from T. kirilowii mitochondria, and the codon usage for each amino acid pair is presented in Table S1. Codons exhibiting a relative synonymous codon usage (RSCU) greater than 1 were deemed to be preferentially utilized by the corresponding amino acids. As depicted in Fig. 3, apart from the RSCU values of 1 for both the start codon AUG and tryptophan (UGG), there exists a prevailing codon usage bias towards mitochondrial PCGs. To understand the codon usage patterns in the mitochondrial genome of T. kirilowii, we analyzed various parameters. Among mitochondrial PCGs, alanine (Ala) has the highest RSCU value of 1.56 and demonstrates a preference for GCU. The stop codon, which prefers UAA, has an RSCU value of 1.54. These genes show a preference for codons ending in either A or T. The third positions of the codons T, C, A, and G have probabilities of 0.3421, 0.2841, 0.3332, and 0.2715, respectively (Table 3). The codon bias index (CAI) is 0.161, the frequency of optimal codons (Fop) is 0.387, the codon bias index (CBI) is −0.039, and the effective number of codons (ENc) value is 58.8. These parameters indicate a well-balanced utilization of all codons in the mitochondrial genome (Table 3).

Figure 3 Analysis of relative synonymous codon usage from T. kirilowii mtDNA.

Table 3 Analysis of codon bias from T. kirilowii mtDNA.

Title	T3s	C3s	A3s	G3s	CAI	Fop	CBI	ENc	
	0.3421	0.2841	0.3332	0.2715	0.161	0.387	−0.039	58.80	

Repeat elements analysis

The analysis revealed a total of 93 simple sequence repeats (SSRs) on Chromosome 1, with monomeric and dimeric forms comprising 61.29% of the SSRs identified. Specifically, thymidine (T) monomeric repeats accounted for 55.26% (21 out of 38) of the monomeric SSRs (Fig. 4A, Table S3). No hexametric simple sequence repeats (SSRs) were detected in the mitochondrial genome under investigation (Fig. 4). Tandem repeat sequences, also referred to as satellite DNA, consist of core repeat units spanning approximately 7 to 200 bases that are extensively repeated consecutively. These repetitive elements are ubiquitously present across eukaryotic and prokaryotic genomes. Ten tandem repeats ranging from 9 to 39 base pairs were identified on Chromosome 1. These repeats had a sequence similarity of over 72%. Additionally, the presence of scattered repeats on Chromosome 1 was investigated. The study found a total of 270 pairs of repeats, each with a length of 30 or more base pairs. Among these, there were 128 pairs of palindromic repeats and an equal number of forward repeats, totaling 256 pairs. No reverse repeats or complementary repeats were detected in this study. The longest observed palindromic repeat was 282 bp, while the longest forward repeat was measured at 138 bp (Fig. 4B, Tables S2 /S4). On Chromosome 2, 34 SSRs were identified, with monomeric and dimeric forms comprising 52.94% of the total SSRs. Among the 12 monomeric SSRs, thymidine (T) repeats accounted for 58.33% (7 out of 12) (Fig. 4A, Table S6). A total of 34 pairs of repeated sequences were found, with a minimum length of 30 base pairs. These repeat pairs included both palindromic (13 pairs) and forward (21 pairs) repeats, but no reverse or complementary repeats were detected. Notably, the longest palindromic repeat spanned 64 base pairs, while the longest forward repeat extended up to 173 base pairs. This information is also presented in Fig. 4B and Tables S5 /S7. The distribution of both types of repeats showed a more balanced pattern across both mitochondrial chromosomes, as shown in Fig. 4C.

Figure 4 Repeat fragments analysis of T. kirilowii mtDNA.

(A) Number of simple sequence repeats (SSRs) in the two chromosomes of T. kirilowii mtDNA. (B) Number of dispersed repeats in the two chromosomes of T. kirilowii mtDNA.C. Chordal graph of repeats in the two chromosomes of T. kirilowii mtDNA. Organe lines indicate palindromic repetition, violet lines indicated forward repeats, and the bars on the second and third circles indicate the presence of tandem repeats and SSRs, respectively.

Identification and validation of RNA-editing sites

The RNA editing events were identified in 38 unique protein-coding genes (PCGs) from the mitochondria of T. kirilowii. The established criterion for identification was a cutoff value of 0.9. Based on this criterion, a total of 518 potential RNA editing sites were identified across 38 mitochondrial protein-coding genes, all of which involved C-to-U base changes (Table S8). Among the mitochondrial genes, a total of 42 RNA editing sites were identified in nad 4, which exhibited the highest frequency of editing events among all mitochondrial genes. Subsequently, ccm B displayed 39 instances of RNA editing. The fewest occurrences were observed in nad 3, rpl 2, and rps 12, each having only one editing event (Fig. 5A). RNA editing events predominantly occurred at the second codon position, accounting for 58%, as compared to 31% at the first codon position and 5% at the third codon position. Interestingly, there are simultaneous editing events at two locations of the same codon (first and second, second and third, first and third), with the highest probability of simultaneous editing occurring at the first and second positions of codons (Fig. 5B). During the 518 editing events, a total of 244 leucine amino acids were produced, making up 47.1% of the total produced amino acids. Phenylalanine accounted for 18% of the total amino acids. Both leucine and phenylalanine are hydrophobic and can contribute to the improved folding of proteins. The editing events resulted in the generation of three stop codons in atp 6, atp 9, and rps 10. In addition, eight new start codons were created, all of which were modified by threonine (as shown in Fig. 5C).

Figure 5 Characteristics of the RNA editing identified in PCGs of T. kirilowii mtDNA.

(A) Count of RNA editing sites in mitochondrial PCGs of T. kirilowii. The number of editing sites included in each gene was annotated above the columns. (B) Number of codon editing sites. (C) Number of RNA sites edited for different types of amino acids.

Horizontal genome transfer from chloroplast to mitochondria

During a sequence similarity analysis, 15 fragments were found to have similarities to the mitochondrial and chloroplast genomes. These fragments had a total length of 23,971 base pairs, which is equivalent to 6.80% of the entire mitochondrial genome. Out of the 15 fragments, 12 were located on chromosome 1, while the remaining three were found on chromosome 2. The longest fragment was 7,392 bp in length, and the shortest fragment was only 52 bp long. Further analysis of the homologous sequences revealed that the 15 fragments were present in 15 complete genes. These genes included eight protein-coding genes (psa B, ndh B, pet G, psa A, psb E, psb F, rpl 23, rps 7) and seven tRNA genes (trn D-GUC, trn H-GUG, trn I-CAU, trn L-CAA, trn N-GUU, trn P-UGG, trn W-CCA). For additional information, please refer to Fig. 6 and Table S9.

Figure 6 Comparative alignment between chloroplast and mitochondria DNA of T. kirilowii shows 15 large syntenic blocks (pale orange).

Phylogenetic analysis

Based on the DNA sequences of 26 conserved mitochondrial PCGs, a phylogenetic tree was constructed for 38 angiosperm species categorized into four groups (Cucurbitales, Rosales, Fagales, Fables). The common protein-coding genes identified in this study included atp 1, atp 4, atp 6, atp 8, ccm B, ccm C, ccm FC, ccm FN, cob, cox 3, mat R, mtt B, nad 1, nad 2, nad 3, nad 4, nad 4L, nad 5, nad 6, nad 7, nad 9, rpl 2, rpl 16, rps 3, rps 4, and rps 19. The two mitochondrial genomes of the Fabales were used as the outgroup for comparison. The mitochondrial DNA-based Phylogeny topology is per the recent classification by the Angiosperm Phylogeny Group (APG). T. kirilowii is classified within the Cucurbit Fales order (Fig. 7).

Figure 7 Phylogenetic relationship of T. kirilowii with related species based on 26 protein-coding genes shared by all the mitochondria DNA.

The bootstrap support value is indicated by the number on each branch, while the families of each species are represented by different colors.

Comparative analysis of five Cucurbitaceae mitochondrial genome

A comparative analysis of the mitochondrial genomes of five species belonging to the Cucurbitaceae family (T. kirilowii, C. lanatus, M. charantia, L. acutangular, H. pedunculosum) was conducted to gain a better understanding of the genetic relationship of T. kirilowii within the family. The mitochondrial genome sizes of these species ranged from 224,863 to 460,333 bp, with L. acutangular having the largest genome and H. pedunculosum having the smallest. The GC content in the coding regions of the mitochondrial genomes varied from 43.87% to 47.12%. T. kirilowii had 61 genes, including 38 protein-coding genes (PCGs), while C. lanatus had 58 genes, including 37 PCGs. M. charantia had 65 genes, including 39 PCGs, L. acutangular had 63 genes, including 40 PCGs, and H. pedunculosum had 54 genes, including 36 PCGs. Interestingly, the largest (L. acutangular) and smallest (H. pedunculosum) mitochondrial genomes differed by only nine genes, which included five transfer RNAs (tRNAs) and four PCGs (Table 4).

Table 4 General characteristics of five Cucurbitaceae mitochondrial genome.

	T. kirilowii	C. lanatus	M.charantia	L. acutangula	H. pedunculosum	
Accession no.	PP625757.1	NC_014043.1	NC_077563.1	NC_050067.1	MW497575.1	
Size (bp)	352,749	379,236	331,440	460,333	224,863	
GC%	45.39	45.61	45.60	43.87	47.12	
Genes	61	58	65	63	54	
tRNAs	20	18	23	20	15	
rRNAs	3	3	3	3	3	
PCGs	38	37	39	40	36	

The relationship between the mitochondrial genomes of T. kirilowii and four other Cucurbitaceae species was evaluated using the BLASTN program to analyze homologous genes and sequence arrangement. As shown in Fig. 8, the red arc area indicates the region where the inversion occurred, while the gray area represents the region exhibiting good homology. To improve the accuracy of the findings, we excluded collinear blocks shorter than 0.5 kb from our analysis. Our study revealed a significant number of homologous collinear blocks between T. kirilowii and other species of Cucurbitaceae. Interestingly, we found certain regions in T. kirilowii that appeared unique and did not show any homology with other species. The arrangement of collinear blocks among the five species was found to be inconsistent. We also found that mitochondrial genomes of T. kirilowii underwent a significant number of genome rearrangements compared to their close relatives. Additionally, the mitochondrial genome sequences of all five Cucurbitaceae species showed highly non-conserved arrangement orders and frequent genome recombination.

Figure 8 Synteny analysis of five Cucurbitaceae species.

The bars represent the mitochondrial genomes, while the red arcs indicate inverted regions. The gray arcs denote better homologous regions. Unique regions in each species are indicated by the absence of colinear blocks.

To further analyze the mitochondrial genomes, the coding sequences were extracted to identify shared genes. Among all five species, 31 shared genes (atp 1, atp 4, atp 6, atp 8, nad 1, nad 2, nad 3, nad 4, nad 5, nad 6, nad 7, nad 9, nad 4L, mat R, mtt B, rps 1, rps 3, rps 4, rps 12, rps 13, rps 19, sdh 3, sdh 4, cox 2, cox 3, rp1 5, rpl 16, ccm B, ccm C, ccm FC, ccm FN) have been found and utilized them to construct a phylogenetic tree (Fig. 9). Interestingly, the results revealed a close sister relationship between T. kirilowii and L. acutangular.

Figure 9 Quantitative analysis of protein coding genes in the mitochondrial genome across five Cucurbitaceae species.

Molecular evolutionary analysis of T. kirilowii versus four Cucurbitaceae species

The set of 31 shared protein-coding genes among the five Cucurbitaceae species was utilized to compute the ratio of non-synonymous to synonymous mutations (Ka/Ks). This calculation allowed us to observe the evolutionary impact of environmental stress on the mitochondrial genome. However, only four shared genes (atp 1, ccm FC, ccm FN, and mat R) had complete Ka/Ks values calculated due to high intergenic conservation. As shown in Fig. 10A, atp 1, and ccm FC only demonstrate a positive preference for T. kirilowii and M. charantia, while ccm FN observed positive interactions between T. kirilowii and H. pedunculosum, H. pedunculosum and L. acutangular, as well as T. kirilowii and L. acutangular. Additionally, mat R shows positive selection across multiple species. nad 1, nad 2, nad 4, nad 5, nad 7, mat R, rps 3, and rps 4 exhibited a high level of nucleic acid variability, with mat R showing the highest degree of variability (Figs. 10B, 10C).

Figure 10 Molecular evolutionary analysis.

The Ka/Ks values (A), nucleotide diversity (B) and theta (per site) from S (C) analysis of T. kirilowii versus four Cucurbitaceae species.

Discussion

With the advancement of next-generation sequence (NGS) and third-generation sequence (TGS) technology, it has become increasingly clear that the plant mitogenome is a dynamically evolving entity. It demonstrates remarkable diversity across plants in terms of genome form, size, and gene content (Bi et al., 2022; Choi & Park, 2021; Diaz-Garcia et al., 2019; Fischer et al., 2022; Wang et al., 2019). This highlights the dynamic nature of plant mitogenomes and their significant variability within the plant kingdom. Previous studies have suggested that the plant mitochondrial genome is predominantly found in a singular circular structure. However, recent evidence indicates that the actual configuration of the plant mitochondrial genome within the cell is highly diverse, potentially existing as multi-branched circular or even linear structures (Bi et al., 2022; Cao et al., 2023; Li et al., 2022; Wang et al., 2024). This suggests a greater complexity in the organization of the plant mitochondrial genome than previously thought.

Currently, there are approximately 10 nuclear genome sequences and about 11 fully assembled mitochondrial genomes of the cucurbit family that have been published in the database (https://www.ncbi.nlm.nih.gov/genome/). Previous research indicates that species within the same genera of Cucurbitaceae and with similar nuclear genome sizes display significant variations in their mitochondrial genome sizes (Ma et al., 2022). Cucumis sativus, a member of the Cucurbitaceae species, possesses the smallest nuclear genome size, approximately 226.2 Mb (Li et al., 2019). In contrast, Benincasa hispida (Thunb.) Cogn., also belonging to the Cucurbitaceae species, has the largest nuclear genome size at around 975.6 Mb (Luo et al., 2023). This represents a significant difference of about four times in genome size between regular Cucumis sativus and Benincasa hispida. Alternatively, the mitochondrial genome size of Cucumis melo is estimated to be 2,400 kb (Rodríguez-Moreno et al., 2011), while that of Herpetospermum pedunculosum is 225 kb (Table 4), indicating a tenfold difference in mitochondrial genome size between the two species. In this study, the first de novo assembly of the T. kirilowii mitogenome was achieved using Illumina and Nanopore sequence technology, thereby establishing a reference for its genetic research. The T. kirilowii mitogenome is composed of two circular molecules, with a combined length of 352,749 bp. These circular molecules are designated as chromosome 1 (245,700 bp) and chromosome 2 (107,049 bp) (Table 1) (Fig. 2). Two circular molecules mitochondrial genomes are not rare in plants. Two circular mitochondrial structures have been observed in Gelsemium elegans (Gardner & Champ.) Benth., Salvia miltiorrhiza Bunge., Chaenomeles speciosa (Sweet) Nakai., and sugarcane. These structures consist of a larger circular and a smaller circular between the two molecules, with a notable difference in genome size of 2-3 times. Simultaneously, the genes carried on the two circular molecules are not identical, indicating that the circular molecules exist independently and both play a crucial role in maintaining the stability of the mitochondrial genome (Cao et al., 2023; Lloyd Evans et al., 2019; Yang et al., 2022a; Yang et al., 2022b; You et al., 2022).

In the mitochondrial genome of T. kirilowii, 61 functional genes have been identified, including 38 distinct PCGs, 20 tRNA genes, and three rRNA genes (Table 2). The number of genes and PCGs in the mitogenome did not exhibit a positive correlation. The number of genes in Cucurbitaceae species ranged from 54 to 65, and the number of PCGs ranged from 36 to 40. However, H. pedunculosum had the lowest number of PCGs, consistent with its possession of the smallest mitogenome in comparison to other species within the family (Table 4). The GC content of the mitochondrial genome in T. kirilowii, as well as in other Cucurbitaceae green plants, was found to be similar (Table 4). This result supports the conclusion that GC content is highly conserved among higher plants.

The reorganization of repeat sequences can lead to gene loss or the presence of multiple copies. However, all PCGs in the T. kirilowii mitogenome are present as single copies, with only the three tRNA genes having multiple copies (Table 2). The duplication of the rps 19 gene appears to be a common occurrence in the mitochondrial genome of Cucurbitaceae species. However, this phenomenon is not observed in the mitochondrial genome of T. kirilowii (Alverson et al., 2010; Niu et al., 2023; Ruang-areerate et al., 2020). Additionally, there are no repeat sequences larger than 300 bp in size within the T. kirilowii mitogenome, which potentially contributes to the stability of its genome structure and gene content (Fig. 4, Tables S3–S8). A recent study found that the mitochondrial genome of L. acutangular contains a significant number of large segment repeats, with sizes ranging from 31 to 5,301 bp. These large repeats could be the reason for the overall larger size of the loofah’s mitochondrial genome. In general, the loss of tRNA genes from mitochondria during evolution is offset by the acquisition of sequences transferred from chloroplasts. Similarly, in the case of the T. kirilowii mitogenome, seven tRNA genes were identified in the sequences transferred from the chloroplast genome, as shown in Fig. 6. This study has found that the complete core genes of the photosystem, namely pet G, psa A, psa B, psb E, and psb F, are transferred to mitochondrial genes (as listed in Table S10). However, it is unclear what their specific functions are. In conclusion, the presence of repetitive and foreign sequences affects the expansion of the mitogenome and can have an impact on certain important functions.

RNA editing is a post-transcriptional process that involves alterations to the nucleotide sequence, including insertions, deletions, and substitutions. These changes lead to variations in genetic information (Edera, Gandini & Sanchez-Puerta, 2018). RNA editing is widespread in angiosperm organelle transcripts, with the majority involving the conversion of cytosine (C) to uracil (U). There are also instances of uracil (U) being edited to cytosine (C), as well as adenosine (A) being edited to hypoxanthine (I) (Kovar et al., 2018; Small et al., 2019). Notably, a total of 518 potential RNA editing sites were identified on 38 mitochondrial PCGs of T. kirilowi i, all involving the conversion of cytosine (C) to uracil (U) (Table S9). Meanwhile, RNA editing is more frequently observed in the second position of a codon and has a higher likelihood of being edited for leucine (Fig. 5). Previous research has suggested that the presence of hydrophilic amino acids is crucial in facilitating protein folding, promoting a favorable folding tendency. Conversely, a decrease in the proportion of hydrophilic amino acids has been linked to an increase in overall protein structure stability (Bentolila, Alfonso & Hanson, 2002).

Furthermore, the phylogenetic relationship of T. kirilowii with representative taxa was analyzed based on the mitochondrial genome information. The resulting phylogenetic tree illustrated a clearly defined taxonomic relationship among the taxa. A total of 38 angiosperm species have been categorized into four groups to conduct evolutionary analysis with mitochondrial genomes (Fig. 7). The complete mt genome sequences are being considered to unravel the phylogenetic relations among these closely related species. It is not surprising that similar clustering patterns were observed in the ML tree constructed from mitochondrial genome sequences as those obtained from conserved genes among nuclear genomes (Guo et al., 2020). In terms of classification, T. kirilowii is more closely related to the loofah in both types (Figs. 7, 9). This finding suggests a significant correlation between the evolutionary relationships of mitochondrial and nuclear genomes, indicating potential co-evolutionary processes at play.

Repeated sequences play a role in mediating genomic rearrangements, leading to significant variation in mitogenomes among different species (Chevigny et al., 2020). The current findings suggest that the majority of Cucurbitaceae mitogenomes consist of a single circular chromosome, except for T. kirilowii. However, the collinearity within the Cucurbitaceae is not as robust as that within the genus Populus (Fig. 8) (Bi et al., 2022), indicating a rich species diversity in Cucurbitaceae.

The Ka/Ks analysis and comparison of genome features with those of other plant mt genomes offer a comprehensive understanding of the evolution of plant mitochondrial genomes (Han et al., 2022). In general, the majority of the findings in this study are consistent with previous reports (Cao et al., 2023; Cheng et al., 2021; Wang et al., 2024). The genes that underwent neutral or negative selections were also identified in Cucurbitaceae. However, atp 1, ccm FC, ccm FN, and mat R in T. kirilowii mtDNA were found to have undergone positive selection compared to other selected species (Fig. 10A). The mat R gene is the only gene that underwent positive selection and have high nucleic acid variability during the evolution (Fig. 10).

This study has important implications for understanding the morphology and genetic mechanisms of mitochondrial genes, as well as biodiversity, functional diversity, and species evolution. However, compared to the continuous development of plastid genomes, there is still a need to explore the mechanisms of mitochondrial evolution.

Conclusion

The study assembled the mitogenome of T. kirilowii, which has a distinctive multi-branched structure consisting of two circular molecules with sizes of 245,700 bp and 107,049 bp. The genome has 38 unique protein-coding genes (PCGs), 20 tRNAs, and 3 rRNAs. There are also many repeats in the mitochondrial genome, including 127 SSRs (simple sequence repeats) and 318 pairs of dispersed repeats. Among these repeats are 141 palindromic repeats, 163 forward repeats, and 14 tandem repeats. The study found 15 fragments totaling 23,971 bp that are similar to both the mitochondrial and chloroplast genomes. These fragments represent 6.80% of the total mitochondrial genome length. The study also revealed that T. kirilowii has a close genetic relationship with L. acutangula through phylogenetic analysis. The mitochondrial genome also has 518 potential RNA editing sites, all of which were found to be C-U edits. These edits tended to result in the conversion of hydrophobic amino acids. The study also found that mat R exhibits a high level of nucleic acid diversity within the Cucurbitaceae family. This study provides a theoretical framework for understanding the evolutionary process and potential utilization of T. kirilowii germplasm resources.

Supplemental Information

Supplemental Information 1 Relative synonymous codon usage for each amino acid in the mitochondrial genome of T. kirilowii

Supplemental Information 2 Repeat sequences in the mitochondrial genome of T. kirilowii

Supplemental Information 3 Repeat sequences in the mitochondrial genome of T. kirilowii

Supplemental Information 4 Repeat sequences in the mitochondrial genome of T. kirilowii

Supplemental Information 5 Repeat sequences in the mitochondrial genome of T. kirilowii

Supplemental Information 6 Repeat sequences in the mitochondrial genome of T. kirilowii

Supplemental Information 7 Repeat sequences in the mitochondrial genome of T. kirilowii

Supplemental Information 8 RNA editing sites in mitochondrial PCGs of T. kirilowi

Supplemental Information 9 The homologous chloroplast DNA fragment in the T. kirilowii mitochondrial genome

Supplemental Information 10 Complete T. kirilowii mitochondrial genome sequence

Supplemental Information 11 Chromosome 1 sequence

Supplemental Information 12 Chromosome 2 sequence

Additional Information and Declarations

Competing Interests

Author Contributions

DNA Deposition

Data Availability

The authors declare there are no competing interests.

Zhuanzhuan Jiang conceived and designed the experiments, performed the experiments, analyzed the data, prepared figures and/or tables, authored or reviewed drafts of the article, and approved the final draft.

Yuhan Chen performed the experiments, analyzed the data, authored or reviewed drafts of the article, and approved the final draft.

Xingyu Zhang performed the experiments, analyzed the data, prepared figures and/or tables, authored or reviewed drafts of the article, and approved the final draft.

Fansong Meng performed the experiments, prepared figures and/or tables, and approved the final draft.

Jinli Chen conceived and designed the experiments, performed the experiments, authored or reviewed drafts of the article, and approved the final draft.

Xu Cheng conceived and designed the experiments, performed the experiments, analyzed the data, prepared figures and/or tables, authored or reviewed drafts of the article, and approved the final draft.

The following information was supplied regarding the deposition of DNA sequences:

GenBank: PP625757.1; PRJNA1119170.

The following information was supplied regarding data availability:

Relevant supplementary information on this article is accessible in the Supplemental Information.

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
