# Peer review of "Assembly and evolutionary analysis of the complete mitochondrial genome of Trichosanthes kirilowii, a traditional Chinese medicinal plant"

_PeerJ, doi:10.7717/peerj.17747_

## Round 0.1 · original submission · Major Revisions

Thanks for your work to PeerJ. Please address these changes and resubmit.

·

Basic reporting

The manuscript is well – written and English language used is understandable. The introduction to the species is informative. The literature used is update and relevant. However, the figure and table need revision. Also, the NGS raw data (both short and long read) need to be submitted in a public repository.
There is also some comments :
Line 113 species name such as Arabidopsis thaliana in Italic. Please check through the manuscript
Line 135 T. kirilowii (should be in Italic)
Line 134 Please revise Circus package should be Rcircos package. Please check again if there is any other errors through the manuscript
Line 282 L. acutangula species (should be in Italic)
Table 4. The accession number inside the table should be use the underscore (NC_)
Line 428-430 Please include information or title for each supplementary Table S2-S7.
Figure 7 and 8. Please include information the accession number T. kirilowii in the figure.

Experimental design

The methodology used is scientifically standard in this field. However, it needs elaboration and clarification. As suggestion below :

Line 99-104 DNA Isolation and sequencing method
There is no information regarding the raw NGS data sequence from Illumina Hiseq and Oxford Nanopore. Suggested the authors Deposited raw data.
Line 108 Why The mitochondrial genome of T. kirilowii was assembled using only long-read data. Please elaborate why the authors not utilize the short-read data ?
Line 149 2.6 Organellar phylogenetical inference
What is the statistical method do you used to create the phylogenetic figure ?. For instance, Maximum likelihood or bayesian method ?. Please state clearly this in the method inside the paragraph.
The authors should inform the reader that both chromosomes were combined for this phylogenetic study. Please revised this method section
Line 270 stated that the authors used 26 mitochondrial PCGs but not stated in the method section. Please include this information.

Validity of the findings

The data provided by the authors support the findings. However, it is a lack of discussion to support the claim such as why T. kirilowii has a double mitogenome.

Additional comments

Please discuss and explain why this species has a double mitogenome. Is it a common phenomenon in eukaryotes or plants ?
Explain why only T. kirilowii have double mitogenome among the Cucurbitaceae family. What is the situation in other families?
Please do a comparative analysis with the other plants that also have a double mitogenome such as sugarcane. Are they have similar structures and gene content on both chromosomes ?, etc. This is much more relevant to explain the situation in plants especially in order to support the findings.

Reviewer 2 ·

Basic reporting

This study reports the complete assembly and annotation of the mitochondrial genome of Trichosanthes kirilowii, a species of the genus Trichosanthes in China. The genome spans 352,749 bp and consists of 2 circular subgenomes of varying sizes, encoding 38 protein-coding genes, 20 tRNAs, 3 rRNAs. The article conducted analyses such as gene horizontal transfer events, phylogenetic relationships, and collinearity analysis of multiple Cucurbitaceae species, further revealing the evolutionary events and processes of this species from a mitochondrial perspective. This laid a solid molecular foundation for subsequent analysis of Cucurbitaceae. However, there are the following issues that need further revision.
1. In the abstract, the phrase "the genus Cucurbitaceae" may be missing an "of" between the family name and the genus name. Please check and revise.
2. The first appearance of a species name should be accompanied by the author's information. Please add this. Additionally, in multiple places throughout the article, such as on lines 113 and 134, species names are not italicized. Please correct this.
3. In the text, there are two formats for referring to figures: "Figure5C" and "Fig5A". The format for "fig" should be consistent.
4. The gene horizontal transfer events lack analysis of transfer between the nuclear genome and the mitochondrial genome. If the nuclear genome has been published, it can be supplemented as appropriate.
5. In Figure 1, "chromosome" should be changed to "scaffold" for better accuracy.
6. In Figures 2 and 9, gene names should be italicized.

Experimental design

no comment

Validity of the findings

The article conducted analyses such as gene horizontal transfer events, phylogenetic relationships, and collinearity analysis of multiple Cucurbitaceae species, further revealing the evolutionary events and processes of this species from a mitochondrial perspective. This laid a solid molecular foundation for subsequent analysis of Cucurbitaceae.

Additional comments

no comment

---

## Round 0.2 · accepted · Accept

Congratulations! Thanks for your work to PeerJ.

·

Basic reporting

The errors have been revised as per my previous comments. The manuscript has been improved.

Experimental design

The authors addresed all my previous comments regarding methods section and deposited the raw data. The manuscript has been improved.

Validity of the findings

The authors addressed all my previous comments regarding the discussion and provided additional information or statements to support the results. The manuscript has been improved.

Reviewer 2 ·

Basic reporting

The manuscript has improved after revision and can be accepted now.

Experimental design

no comment

Validity of the findings

no comment

Additional comments

no comment